# Use of Cloud-Based Virtual Reality in Chinese Glove Puppetry to Preserve Intangible Cultural Heritage

Der-Lor Way [1,*] and Yu-Hsien Wei [2]

1    Department of New Media Art, Taipei National University of the Arts, Taipei 112, Taiwan
2    Department of Computer Science and Engineering, National Sun Yat-sen University, Kaohsiung 804, Taiwan; joewei0226@gmail.com
*    Correspondence: adler@newmedia.tnua.edu.tw

**Abstract:** Chinese traditional glove puppetry is a folk art with a long history. It is worth inheriting and safeguarding this distinguished intangible cultural traditional art using virtual reality. With this background, this study integrates the digital resources of glove puppetry from the perspective of satisfying users' performance needs. In this study, a multi-user, cloud-based virtual reality glove puppetry system was developed that enhances the classic works of glove puppetry. Each user has a unique perception of the virtual environment and can interact remotely. The system involves human–computer and human–human interactions. This study also describes the design and control of glove puppets. The virtual reality system provides a unique entertainment experience to users of all ages. Through a questionnaire administered to 30 subjects after the user play, this study investigated the operation and experience of this system. According to the research findings, the proposed cloud-based VR system is not only easy to use, but also helps to preserve traditional intangible culture. Our research has high theoretical value and can help preserve traditional glove puppetry. Our cloud-based virtual reality system offers a new application for disseminating and preserving intangible cultural heritage.

**Keywords:** intangible cultural heritage; glove puppetry; virtual reality; human–computer interaction; cloud network

## 1. Introduction

Forms of intangible cultural heritage include practices, knowledge, skills, representations, objects, artifacts, instruments, communities, and cultural spaces associated therewith. Intangible cultural heritage is passed down through generations, constantly reinvented by communities in response to changes in their environment and history. Intangible cultural heritage affords communities with a sense of identity and stability and is a key aspect of cultural diversity and human creativity [1]. The "Convention for the Safeguarding" of the Intangible Cultural Heritage treaty proposed by the UNESCO (United Nations Educational, Scientific, and Cultural Organization) divides intangible cultural heritage into (1) oral traditions; (2) performance art; (3) social customs, etiquette, and festivals; (4) knowledge and practices about the environment; and (5) traditional crafts.

Intangible cultural heritage is more challenging to safeguard than tangible cultural heritage because performance art, music, and crafts have no distinct form. The conservation of intangible heritage is also complicated by its long history [2]. In addition, the COVID-19 pandemic in 2020 created new challenges for exhibition centers and apprenticeship training and prevented festivals about intangible heritage from being held, necessitating the creation of new methods of preservation and civic tourism [3]. However, the COVID-19 pandemic also created challenges for museums. Because museums were forced to close during the pandemic, they began experimenting with new methods of operating, such as online programs [4,5]. The online use of the museum not only created digital exhibitions, but also expanded the exhibition space. Virtual reality (VR) and augmented reality (AR)

can enhance museum visitors' experiences [6]. Museums are exploring the use of digital content for exhibitions and experiences by conducting preference studies [7]. Considering these features, the use of VR and AR has increased in museum exhibitions involving intangible heritage.

In the digital age, which pursues sophistication and speed, traditional glove puppetry struggles to attract the attention of people. To inherit the legacy of traditional culture, many declining traditional skills have brought the younger generation's understanding of our country's cultural history through emerging technology and application. VR is an emerging technology that facilitates effective learning, communication, and entertainment within cultural spaces. The virtual displaying of cultural artifacts may involve interactive storytelling. Captivating storylines within virtually reconstructed three-dimensional (3D) environments can engage consumers and provide both educational and entertainment value. AR and VR can play key roles in presenting past and cultural practices through innovative methods [8]. Physical movement in VR can be an effective learning activity because it can be more motivating and engaging than traditional teaching methods [9,10]. This means that virtual reality can provide a superior understanding of the content of intangible cultural heritage.

The purpose of this study was to use cloud-based virtual reality in Chinese glove puppetry to preserve intangible cultural heritage. Therefore, this study developed a multi-user, cloud-based virtual reality glove puppetry system that enhances the classic works of glove puppetry, as shown in Figure 1. Each user has a unique perception of the virtual environment and can interact remotely. In addition, the Internet enables users to collaborate in different locations to perform together. To ensure that collaboration in VR is effective, participants' gestures and emotions must be captured and transmitted in real-time [11]. Therefore, teamwork allows players to work collaboratively in different places. Finally, a user experience study is included in this article. User experience research focuses on developing a system usability scale and provides an understanding of puppet cultural preservation. The main contributions of this study are as follows:

1. This paper presents a collaborative work of a puppetry show in a multi-user virtual reality system.
2. The user experience evaluation was implemented to identify predictive and interpretive use measures for these technologies.
3. The proposed cloud-based VR system is easy to use and is not limited to users of a particular group.
4. Our VR system not only provides experience of the operation puppetry but also helps preserve traditional intangible culture.

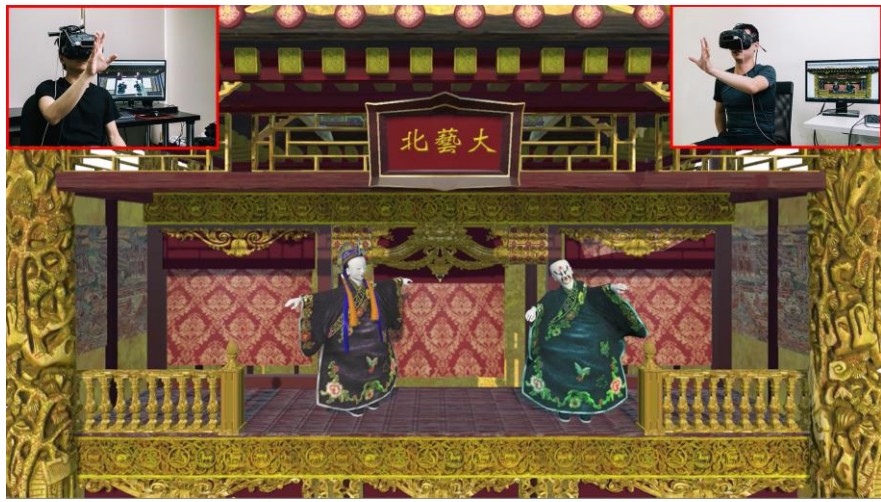

**Figure 1.** Chinese glove puppetry in cloud-based VR.

The remainder of this paper is arranged as follows. Section 2 reviews literature research. Section 3 describes glove puppetry digital design including stage and puppets. Section 4 presents the proposed cloud-based virtual reality system. A user experience study was implemented and is discussed in Section 5. Finally, Section 6 concludes the paper and provides suggestions for future work.

## 2. Literature Research

### 2.1. Chinese Glove Puppetry

Chinese glove puppetry incorporates performing art, visual art, music, classical literature, and folklore. The UNESCO included Chinese glove puppetry in its list of intangible cultural heritage in "Convention for the Safeguarding" in 2012 [12]. Glove puppetry is a type of performance in which puppets animate and create characters, stories, and worlds. Glove puppetry originated in China's Fujian province in the 17th century. Glove puppetry is not only an integral part of folk art, but also a long history of traditional art. Glove puppetry began as street entertainment and was introduced to Taiwan in the mid-19th century [13]. Glove puppetry has since become a popular art form in Taiwan and is often performed during celebrations, such as weddings and religious ceremonies [14]. Glove puppets have deeply influenced the daily life of agricultural societies. They often spread morality and social order, serving as beliefs in the Confucian spirit through the storytelling of a glove puppet show. In the 1950s, Golden Ray's puppets revolutionized the art form. The introduction of television in the 1960s enabled the art forms to develop further. Puppet shows were broadcast on Taiwanese television, and their popularity skyrocketed in the proceeding decade [15]. With the transition from an agricultural society to an industrial society, traditional glove puppetry face challenges such as shortages of trainees, audiences, and funds to preserve their heritage.

Generally, there are six typical roles: male protagonists, female protagonists, painted faces, elders, clowns, and monsters. Each character's makeup, facial expressions, clothing, props, and gestures express its characteristics [16], as shown in Figure 2. For example, a male protagonist's exquisite facial features and graceful clothes show that he is a garbed young male scholar. A painted face displays his confident, courageous, and adventurous warrior character. In traditional glove puppets, such archetypes and attributes are crucial because they allow the audience to quickly identify each puppet's role in the play. Traditionally, glove puppets have heads, hands, and feet made of wood. The puppeteer carefully carves a piece of wood into the shape of a hollow human head, which is then refined and polished. The most embroidered dresses and beautiful props are cloth decorated with absorbing tassels, beads, treasures, feathers, etc. The beautiful craftsmanship of glove puppets often attracts audiences.

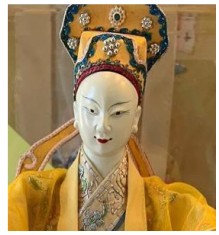 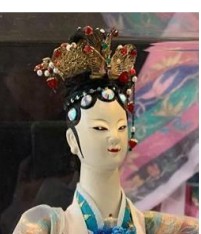 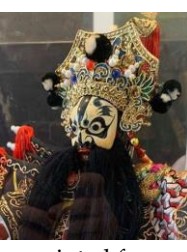 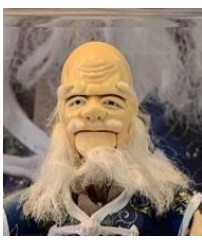 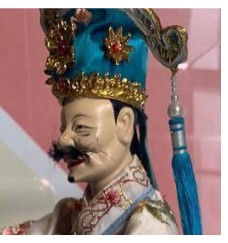 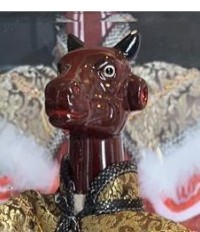

| male role | female role | painted face | elder | clown | monster |

**Figure 2.** Six typical character roles in puppetry Glove puppets [16].

Puppeteers must be highly skilled in manipulating puppets using their fingers. A puppeteer inserts his index finger into the hollow center of the puppet's head; his thumb controls one hand of the puppet; and the other three fingers hold the other hand of the puppet. Puppeteers bring puppets to life by skillfully manipulating their movements and gestures, as shown in Figure 3. Glove puppets have hollow wooden heads that are carved in the shape of a human head. The torso and limbs of the puppets are made of cloth. A

hand enters the glove puppet costume and performs with it during the show. Moreover, glove puppetry performances involve sets and backcourt music. The scripts are passed down orally by storytellers and are typically adapted from classic novels such as Journey to the West, Romance of the Three Kingdoms, Water Margin, and The Seal of the Gods. Unfortunately, only a few traditional glove puppet shows have survived [13].

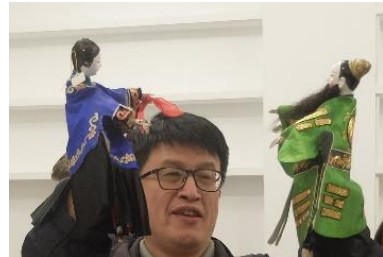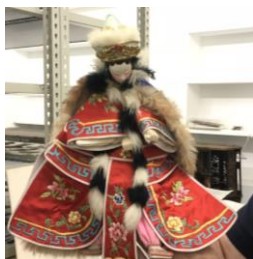

**Figure 3.** Glove puppets.

## 2.2. Virtual Reality and Cultural Heritage

Recently, proponents of cultural heritage are increasingly using VR in museum exhibitions. These technologies can create user-centric experiences and produce cultural heritage digitally accessible, especially when physical admission is constrained. Various methods of using VR to protect intangible cultural heritage have been proposed. Skublewska-Paszkowska et al. gathered many papers about 3D technologies for intangible cultural heritage preservation [17]. They investigated the use of 3D technologies to maintain intangible cultural heritage and discovered that intangible cultural heritage is a necessary extension of tangible cultural heritage. The growing attentiveness of intangible cultural heritage in scientific studies is noticeable. Societies have come to understand the cost of losing their intangible cultural heritage. Research on the use of 3D technologies such as 3D scanning, 3D modeling, VR, and AR to reproduce cultural heritage has increased. Zhao proposed a digital method for preserving intangible cultural heritage [2], including augmented reality technology. They examined the main characteristics of intangible cultural heritage, as well as the artifacts and practices of a culture that have been preserved and transmitted throughout generations. Their AR can greatly improve user experience.

Further, Day and Way established a VR program for a white crane folk dance performed in Mazu temples in Taiwan [18], as shown in Figure 4. In the first phase, the Kinect module was used to capture the performance of the folk dancers and animate the motion of the 3D animation characters (white crane). In the second phase, a Kinect and HTC (High Tech Computer Corporation) VIVE tracker were used to capture user actions. Using the Unity integration platform, the 3D animation characters played, and the 3D avatar to be instantly controlled was presented to the user in the environment. A VR headset was paired with the Kinect motion capture sensor to immerse users into a 3D virtual world where they could view the dancing cranes and appreciate a dynamic virtual environment in real time.

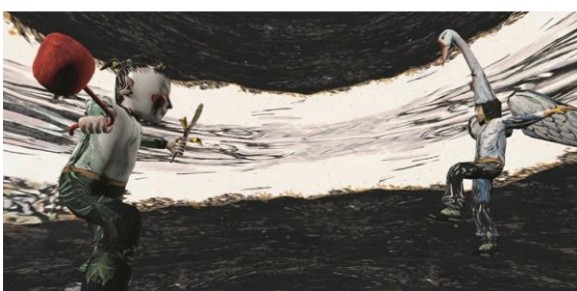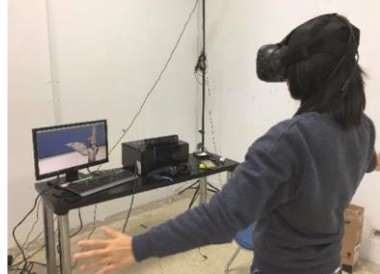

**Figure 4.** White crane folk dance performed in VR [18].

Selmanovic et al. improved user-friendliness in relation to intangible cultural heritage protection using virtual reality [19]. They investigated whether a VR-based 360-degree video could facilitate immersion in applications for the safeguarding of intangible cultural heritage and thereby increase its accessibility. Huang et al. provided the self-issuance method to the folk intangible cultural heritage center [20]. They presented a method of presenting intangible cultural artifacts by using VR headsets. Unlike traditional AR techniques, this approach enables users easily and fully to interact with intangible cultural heritage in an immersive virtual environment. Han and Yi investigated how digital exhibitions of intangible heritage were used during COVID-19 to help audiences understand intangible heritage [21]. They also explored how museums can use digital technology to reproduce intangible heritage for in-person experiences.

Nitsche and McBride explored controlling puppets in VR and puppet-based VR games [22]. Through their bottom-up design approach, they created a VR game in which users adopt the puppets perspective, thereby demonstrating the various design possibilities that VR offers. Lin et al. proposed a method for controlling a virtual puppet by using a computer [23]. Users can select from different puppets, scenes, and music. They used Leap motion controllers to capture the user's gestures and display the puppet's movements on a computer screen. Unfortunately, they only build a 3D scene with a fixed viewpoint. In fact, their system is a 3D multimedia system without immersion. So far, some above virtual reality systems for a single user have been established for intangible cultural heritage. No multi-user, cloud-based VR systems have been developed to present intangible cultural heritage. For this reason, this paper proposed a multi-user, cloud-based VR glove puppetry system to protect intangible cultural heritage.

### 2.3. Virtual Reality Collaboration Systems

VR collaboration systems are becoming increasingly popular for their ability to enhance collaboration through multi-user functionality. Interest and research in this technology have increased, with organizations seeking to capitalize on this. Researchers have explored aspects of remote collaboration that can facilitate communication and expression. Thomas proposed a low-cost, lightweight, easily configurable MuVR device (multi-user virtual reality) [24]. Their MuVR system is portable and without the requirement of a high-priced external infrastructure. Nguyen et al. established a CollaVR [25]: an application that enables multiple users to collaborate and review a VR video together through a network. Ibayashi et al. created a Dollhouse VR [26], a multi-user VR workspace for collaborative design. Piumsomboonet et al. delivered a multi-user VR and AR platform for remote collaboration [27]. Sra et al. provided a shared VR experience for remote users known as "Your Place and My Place" [28].

Some of the above-mentioned VR systems have been held back by poor accessibility and lack intuitive multi-user capabilities. MuVR is portable but has a low resolution. CollaVR is a video-based VR that is realistic but lacks interaction. Although Dollhouse VR and CoVAR allow users to collaborate on spatial tasks in a shared space, they are not suitable for cultural heritage applications. In fact, the system design and requirements are different depending on the application. Most VR applications require trained people to perform well in a virtual reality environment. Therefore, it is important that user experience studies evaluate system usability and requirements. Unfortunately, most of the aforementioned systems do not conduct user experience studies.

In summary, multi-user virtual reality is an emerging technology and a great tool for facilitating collaboration. For collaborative tasks in virtual reality systems, the gestures and sensations of immersed users must be accurately exchanged and visualized in real time. By using a VR periphery network, we developed a multi-user, cloud-based VR glove puppetry system in which users in different locations could perform together. Finally, our user study shows that our VR system is not only easy to use, but also helps preserve the traditional glove puppetry culture.

### 3. Glove Puppetry Design

As mentioned before, a puppet's head is a hollow carved wood with a human-like head. The puppet's body is composed of draped fabric clothes. A hand goes into the glove puppet costume for a performance. Moreover, the stage for the puppetry is the frontcourt in the puppet show. Over time, the design of glove puppetry stages has changed along with the values and customs of society. Puppet theaters were typically found in temples, town squares, and marketplaces. There was no fixed stage in the early period of puppetry. Shows could be staged on a simple and straightforward stage. A portable puppet stage known as the "four-corner shed" was invented in the Qing Dynasty. Later on, the stage design was modified into a colorful hexagonal shed.

### 3.1. Stage Design

Our digital stage was inspired by the stage of the Seden Puppet Theater Foundation, located in the library of Taipei National University of the Arts [29] (Figure 5a,b). This hexagonal shed is adorned with ornate gold leaf and intricately painted maps, creating resemblance to a temple. The stage consists of three parts: the upper cover, base, and screen column. The upper cover acts as the roof and is supported by dragon-shaped pillars, and the base contains window frames. The stage is divided into two floors, with three doors and three windows on each floor. A plaque is inlaid into the upper floor, featuring the name of the school elegantly engraved. Curtains are hung across each door, and the words "General" and "Chancellor" are displayed on the lower sections of the curtains. The base is as tall as the puppeteer's chest. A wooden frame beneath the base provides structural support. According to the above breakdown, Figure 6 displays our 3D stage design for a cloud-based VR puppet show.

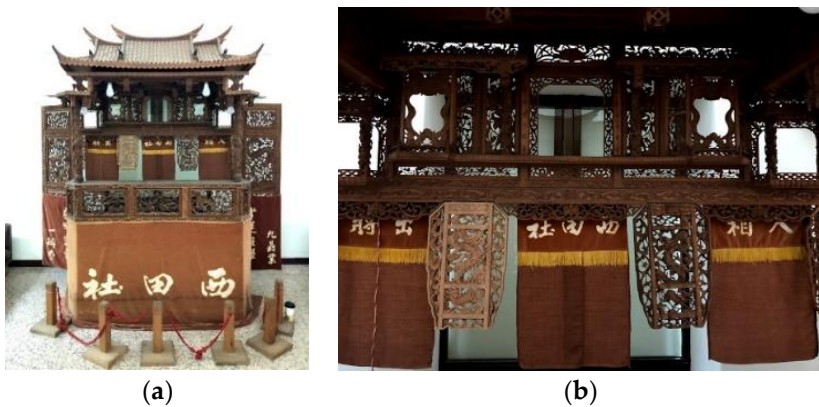

| (**a**) | (**b**) |

**Figure 5.** Seden puppet theater. (**a**) Seden puppet theatre. (**b**) Details of Seden puppet theater.

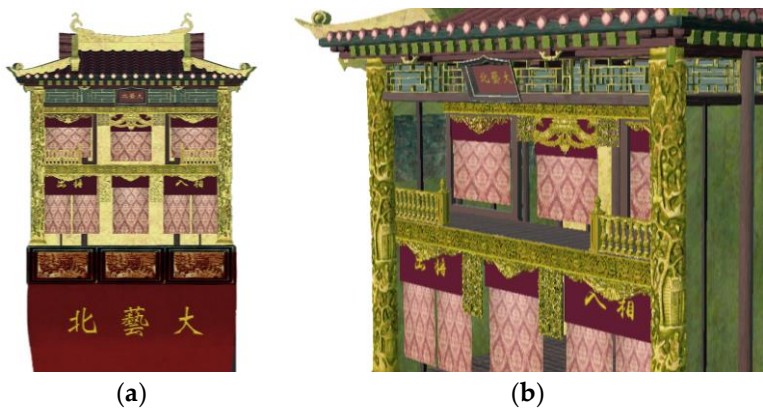

| (**a**) | (**b**) |

**Figure 6.** Stage design for cloud-based VR puppet show. (**a**) Our stage design. (**b**) Details of our stage design.

### 3.2. Glove Puppet Design

Categories of glove puppet characters include the male protagonist, female protagonist, painted face, clown, child, beast, and other. Heroic characters tend to have striking features, with high foreheads and a majestic appearance. Villains tend to have bushy eyebrows, deep-set eyes, pronounced cheekbones, and large mouths. Figure 7 presents a male protagonist and a painted face. Male protagonists are often handsome and enchanting characters. The facial expressions and colors of the characters indicate personality traits, such as loyalty, treachery, kindness, clown, evil, etc.

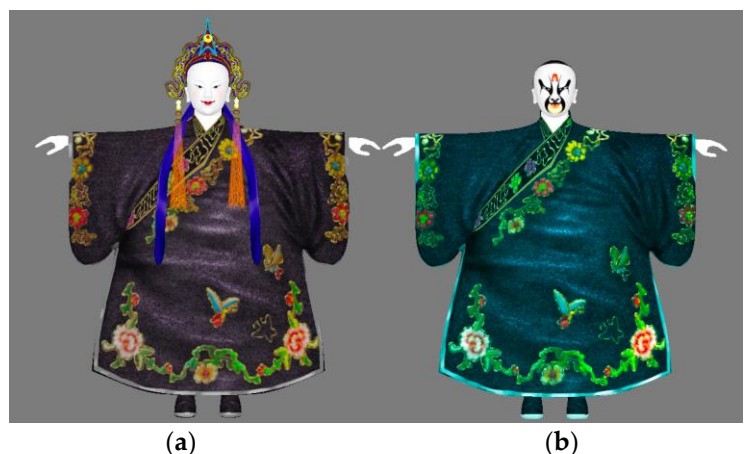

(**a**)            (**b**)

**Figure 7.** Glove puppet design. (**a**) Male protagonist. (**b**) Painted faces.

### 3.3. Clothing Simulation

The personality traits of the characters are also expressed through their different costumes. Simulating the dynamic movements of clothing to enhance the motion of a glove puppet is essential for an effective puppet show. The puppet is controlled by each of the fingers. The puppeteer's gestures are captured using a Leap Motion controller and transferred to the puppet's head and hands. Puppet costumes have three components: the sleeves, the front of the skirt, and the back of the skirt. The clothes of the glove puppet are held up by the fingers, and the texture inside is enhanced.

The dynamic joints were used to simulate realistic movement. The structure of the skeleton and the dynamic joint chains are presented in Figure 8. The hands are controlled by dynamic joint chains. The joints allow for rotation and translation between connected, rigid bodies. Different joints, such as rotary joints, ball joints, online point joints, and pulley joints, can be used to combine rigid bodies into a complete puppet. By defining the degrees of freedom of each joint, the puppet's motion can be constrained. The front and back of the skirt are controlled by five dynamic joint chains each, allowing the costume to swing up and down as the puppeteer moves their hands.

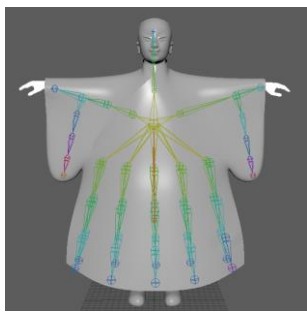 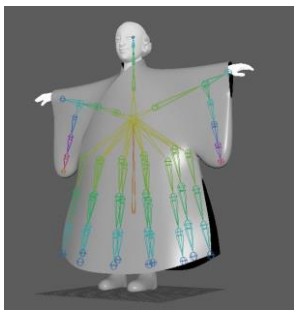 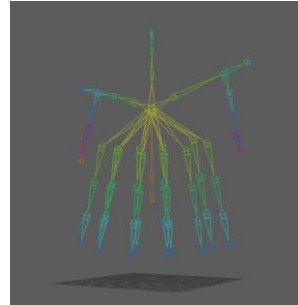

**Figure 8.** Structure of skeleton and dynamic joint chain of glove puppet.

## 4. Cloud-Based Virtual Reality System

This study generates a collaborative, cloud-based virtual reality system for glove puppetry, which is projected to perform a real-time interactive show for safeguarding intangible cultural heritage. The cloud-based VR system we developed is based on Unity. A Leap Motion controller is connected to an HTC Vive head-mounted display, and 3D data are imported into the system. We also improved core technologies and incorporated multi-user functionality into the system. Puppeteers can see each other and the movement of their characters to imitate the interactions with the virtual object.

The schema of virtual reality glove puppetry system is illustrated in Figure 9. When a user registers as a host, the server adds the user and creates a left-hand or right-hand puppet avatar in the scene. The user can control the puppet with their hand through separately acquired data. The Leap Motion controller connected to the HTC Vive VR headset captures the motion of the user's hand. The motion data are transmitted to the computer through the VRPN (VR periphery network). Figure 10 illustrates a user's index finger controlling the puppet's head; the user's thumb and little finger control the puppet's right and left hands, respectively. Two players can join in person, and additional players can join remotely over the Internet. The VR environment is displayed in real time on the connected computer monitor. So, each player can meet and see scenes performed synchronously on their own computer.

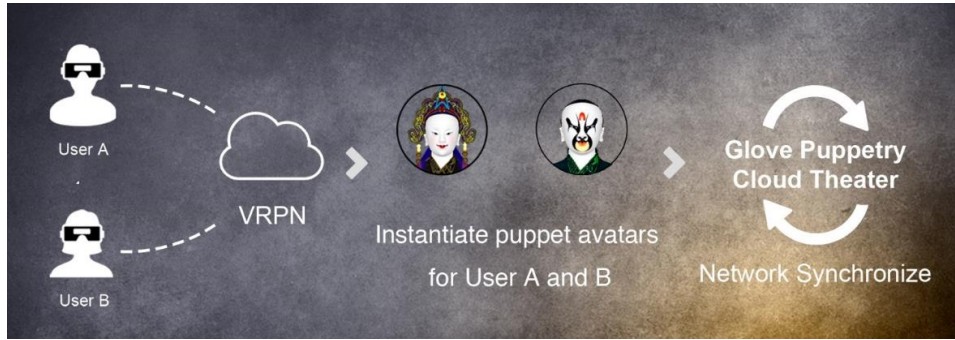

**Figure 9.** Schema of virtual reality glove puppetry system.

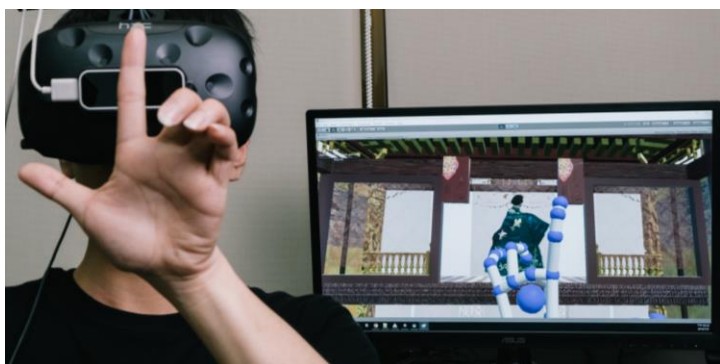

**Figure 10.** Puppeteer wearing HTC head-mounted display and controlling glove puppet avatar.

Data in our system are obtained from Leap Motion controllers and transmitted through the VR peripheral network. The system architecture of the VR glove puppetry system is presented in Figure 11. The network conversion component handles data conversion across the network. The process flow is as follows:

1. Users A and B: Users log into the system. Physical user data are obtained from sensors. For instance, the HTC Vive VR headset catches data on the user's situation and direction, and the Leap Motion controller obtains data on the user's hand. The Leap Rig transfers the data between the sensor and Unity.

2. VR periphery network registration: The Unity network architecture is used to create a client–server network for the VR periphery network. A network manager is spawned to handle network events.

3. Puppet avatars: Three tasks ("Network Identity," "Enable Ctrl," and "CCDIK") are generated during the prefabrication of a glove puppet avatar. Network Identity assigns each avatar a unique ID. Enable Ctrl adds "Avatar Controller" to enable animation when the native player starts. CCDIK (Cyclic Coordinate Descent Inverse Kinematics) handles inverse kinematics for custom skeletons.

4. Avatar controller and dynamic joint: The glove puppet avatar's head and hands are controlled by the user's thumb and index and little fingers, respectively. The avatar controller transfers captured user motion to the avatar. Dynamic joints are attached to the hand bones on the avatar's clothing to make the simulation more realistic.

5. Network transform and network skeleton: In the Unity network architecture, the client cannot communicate directly. Synchronization can only be achieved by the server. The network transform synchronizes the movement and orientation of a player's hands and head. The network skeleton synchronizes the skeleton data of each puppet.

6. Data synchronization: Each client transmits data to the server, and the server transmits data to other clients synchronously.

7. Puppet cloud theater: Puppet performances with users in different locations.

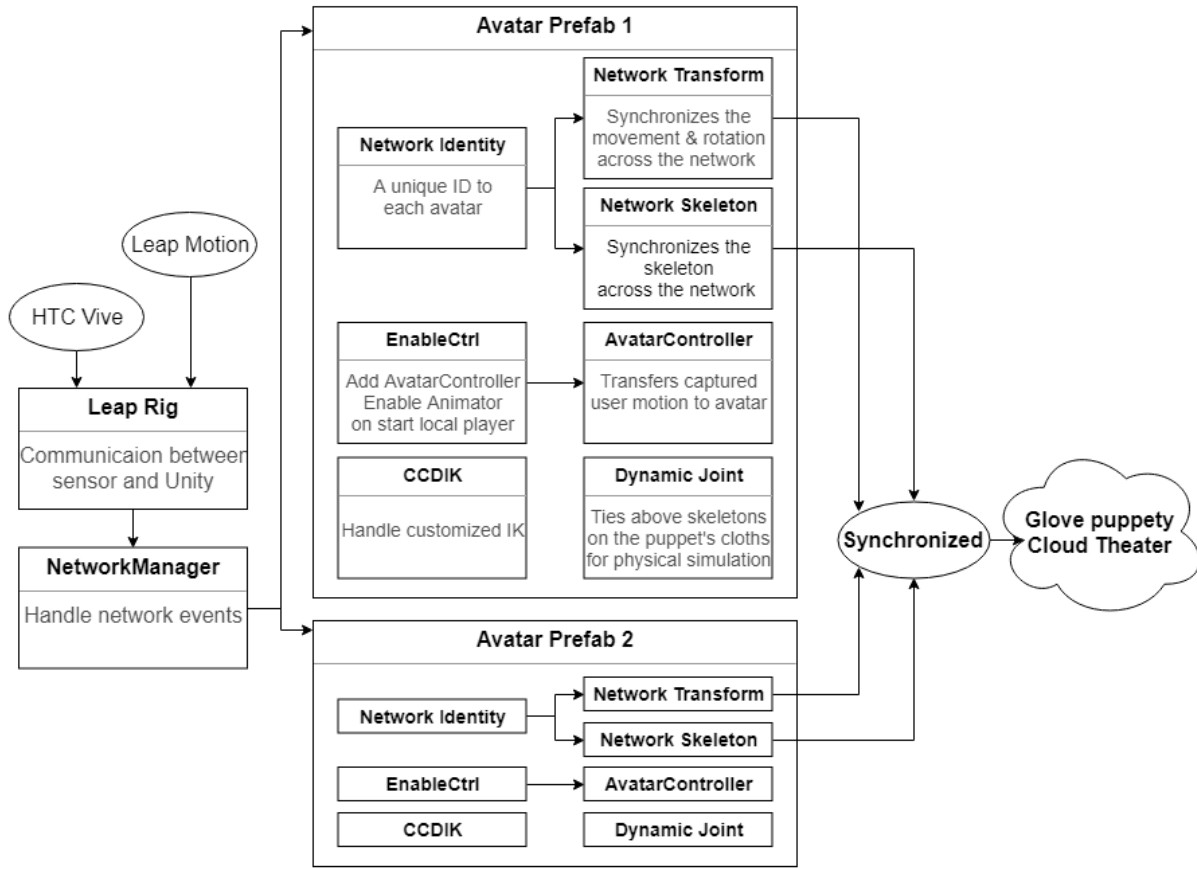

**Figure 11.** System architecture of VR glove puppetry system.

This paper proposed a multi-user, cloud-based VR system which provides different users with the same virtual scene from their own computers. The system enables users to enter the virtual environment by using a computer. Each user has a unique perspective on the virtual scenes. To communicate with others in the virtual environment, users must consider each other's perspectives and experiences of the scene.

## 5. User Experience Study

This article presents a study of user experience. In general, the system usability scale [30] is a common reference for usability surveys in case studies for evaluating user experience design. This usability study focuses on conducting user experience research and provides an understanding of the feasible design methodology included in the Chinese puppet cultural preservation VR project. A preliminary study was conducted to evaluate the cloud-based VR prototypes. A group of 30 participants participated in this user experience study, including 14 females and 16 males ranging in age from 21 to 48 years old. This user experience study focuses on measuring the ease of use and usefulness of the proposed VR system.

The test time for this experiment was approximately 30 min to complete the task. First, the experimental conductor explained the wearing method and operation of the equipment with a demonstration. The purpose and process of the experiment was also described in detail, so that the participants could understand the experiment. Each participant wore a head-mounted display, as shown in Figure 12. The experimental process was notified and to be suspended immediately if there was any doubt or discomfort during the experiment. Subjects were invited to provide a questionnaire referring to their experience of using the system after the end of the experiment, which included three parts: basic information, the system usability scale, and glove puppet culture preservation. There were ten questions on the system usability scale, and five questions on the glove puppet culture preservation satisfaction scale. The question was designed using a five-point Likert scale, with 1 point indicating strongly disagree, 3 indicating no opinion, and 5 indicating strongly agree. Participants can also deliver suggestions that they want to provide to this study for the subsequent adjustment of the system.

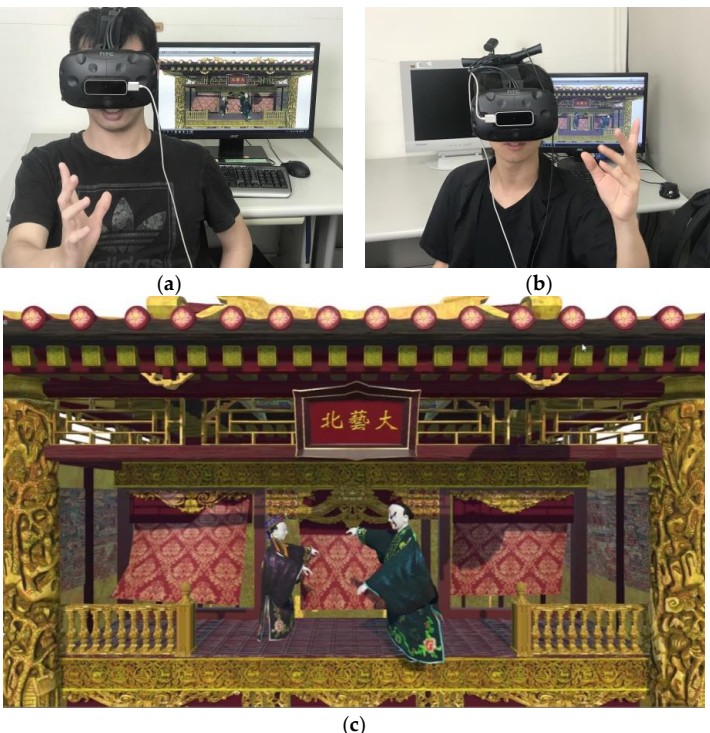

**Figure 12.** Users experience study sharing the virtual stage of our system. (**a**) User A from place A. (**b**) User B from place B. (**c**) Virtual stage.

The Table 1 questionnaire is the System Usability Scale (SUS) using the Likert five-point scale [30]. SUS produces a single number signifying a measure of the whole usability of the system. SUS scores for individual items are not meaningful. To estimate the SUS score, first sum the score of each item's contribution. The singular in Table 1 is a positive

question and the even number is a negative question design. Each item's contribution score will range from 0 to 4. The positive question score is the scale position minus one, and the negative question score is five minus the scale position. The sum of the scores minus 2.5 gives us the overall value of the system usability. The average score is 68 points, which was used as the passing score standard for the system.

**Table 1.** Questionnaire for the system usability scale [30].

1. I think that I would like to use this system frequently.
2. I found the system unnecessarily complex.
3. I thought the system was easy to use.
4. I think that I would need the support of a technical person to be able to use this system.
5. I found the various functions in this system were well integrated.
6. I thought there was too much inconsistency in this system.
7. I would imagine that most people would learn to use this system very quickly.
8. I found the system very cumbersome to use.
9. I felt very confident using the system.
10. I needed to learn a lot of things before I could get going with this system.

Table 2 shows the results of the cloud-based VR system usability statistics table. The average SUS is 82.08 points, which is higher than the average score of 68 points. In other words, our cloud-based VR system is easy to use and not limited to users of a particular group.

**Table 2.** Result of the system usability scale.

|  | Strongly Disagree | Disagree | Neutral | Agree | Strongly Agree | Score |
|---|---|---|---|---|---|---|
| Q1 | 0 | 2 | 3 | 8 | 17 | 3.33 |
| Q2 | 12 | 12 | 4 | 2 | 0 | 3.13 |
| Q3 | 0 | 1 | 4 | 9 | 16 | 3.33 |
| Q4 | 11 | 8 | 7 | 3 | 1 | 2.83 |
| Q5 | 0 | 0 | 5 | 9 | 16 | 3.37 |
| Q6 | 12 | 13 | 4 | 1 | 0 | 3.20 |
| Q7 | 0 | 0 | 3 | 9 | 18 | 3.50 |
| Q8 | 20 | 5 | 4 | 1 | 0 | 3.47 |
| Q9 | 0 | 0 | 5 | 7 | 18 | 3.43 |
| Q10 | 15 | 8 | 6 | 1 | 0 | 3.23 |

In addition to the survey of system usability, the design of items in Table 3 focused on the following research purposes: (1) whether there is new cultural experience or knowledge learning in the cloud-based VR system; (2) the younger generation's understanding of traditional puppetry and the value of traditional cultural heritage; (3) the experience of two people in a collaborative way to perform a puppet show; (4) the feasibility of the Chinese puppet culture preservation using our cloud-based VR system. Table 4 shows the results of the survey questions on Chinese puppet culture preservation. The scores for all the items were above 4.3. The proposed cloud-based VR system not only provides learning and experience of the operation and performance of puppetry, but also helps to preserve traditional intangible culture. Specially, cloud-based VR brings new cultural experiences or knowledge learning to the post-COVID-19 era.

**Table 3.** Questionnaire for glove puppet culture preservation.

| | |
|---|---|
| 11. | I approached the traditional culture of glove puppetry through the cloud-based VR. |
| 12. | I can play glove puppetry with another player through remote collaboration that can facilitate communication and expression to play a show. |
| 13. | The content and graphics of the glove puppetry cloud-based VR are same as the traditional glove puppetry. |
| 14. | Enhance younger generations' understanding of the value of traditional glove puppetry culture. |
| 15. | There are new cultural experiences or knowledge learning with the cloud-based VR in the post COVID-19 era. |

**Table 4.** Result of the Chinese puppet culture preservation.

| | Strongly Disagree | Disagree | Neutral | Agree | Strongly Agree | Mean |
|---|---|---|---|---|---|---|
| Q11 | 0 | 0 | 2 | 9 | 19 | 4.57 |
| Q12 | 0 | 1 | 4 | 8 | 17 | 4.37 |
| Q13 | 0 | 1 | 5 | 7 | 17 | 4.33 |
| Q14 | 0 | 0 | 3 | 9 | 18 | 4.50 |
| Q15 | 0 | 0 | 2 | 10 | 18 | 4.53 |

Furthermore, we asked participants for their comments. On the positive side, users said that "The technology is very interesting, fancy and impressive."; "It gives me a good visual immersive experience."; and "It is great for people new to VR. It has been a great experience". Users also indicated that it "Lacks the tactile feel of puppet physical objects."; that there was a "Problem of force feedback during puppet fighting."; that" Although there are dialogues, it would be better to play sound effects or background music.", and that "The user would like to learn from an expert puppeteer." Occasionally, the quality and stability of the network have an influence on the user experience. Generally, most network VR studies encountered challenges in terms of the server stability.

## 6. Conclusions and Future Work

It is more challenging to safeguard intangible cultural heritage than tangible cultural heritage because performance art, music, and practices have no distinct form. This paper describes the requirements for interactive experiences, digitization of glove puppetry, and a VR glove puppetry system. Our glove puppetry design was inspired by Chinese glove puppetry history, which includes one stage and many glove puppets. In the process of implementation, a set of operational procedures for comprising modules, data integration, and technology implementation of virtual reality networks are planned for traditional Chinese glove puppetry. The system enables puppetry to transcend time and space and become a dynamic interactive experience. Users without professional training can control glove puppet avatars using their hands. This system involves human–computer and human–human interactions and provides an entertaining experience for users of all ages. The questionnaire design of this study includes the system usability scale and the Chinese puppet culture preservation satisfaction scale. After analysis and statistics, the usability score of this system was 82.08 points, which is higher than the average score of 68 points. This demonstrates that the proposed system has good usability and is easy to use. The satisfaction scores for Chinese puppet culture preservation are high, and our system achieves knowledge sharing and learning regarding cultural inheritance. The system developed in this study contributes substantially to the preservation of glove puppetry.

According to participants' suggestions, some functions need to be enhanced in the future. For action feedback, many subjects were expected to collide in regard to the penetration feedback. Collision detection includes collisions between puppets and between puppets and the background. In addition to the dialogue with each other between users,

there should be sound effects or background music. The sound effects showed that the puppet's performance and experience could adhere to reality. However, the users' experience cannot be too long because of hand fatigue. Moreover, the subjects were expected to learn from an expert's puppet skills using our system. In the future, we will provide a performance record system for a grandmaster's puppet.

Looking into the practical applications of the future, the VR glove puppetry system and similar systems can be used to enhance education and increase the accessibility of art forms to the public. VR research also provides the interaction and design for future intergenerational communication for cultural heritage. This system can also be used to promote intangible cultural heritage through interactive VR experiences. It can be used in exhibitions to display intangible cultural heritage and can be promoted based on numbers to promote the popularization of intangible cultural heritage. Intergenerational interaction and learning using our VR system could lead to an exchange between experience and knowledge of traditional culture. The findings of this study can inform the preservation of this form of traditional art. Thus, cloud-based virtual reality has become accessible and reasonable. Moreover, this study's understandings and accomplishments will deliver and relocate this traditional art form, of which we have given a new application.

**Author Contributions:** Conceptualization, D.-L.W.; methodology, D.-L.W. and Y.-H.W.; software, Y.-H.W.; validation, Y.-H.W. and D.-L.W.; writing—original draft preparation, D.-L.W. and Y.-H.W.; Writing—review & editing, D.-L.W. and Y.-H.W.; supervision, D.-L.W. All authors have read and agreed to the published version of the manuscript.

**Funding:** The authors would like to thank the Ministry of Science and Technology of the Republic of China, Taiwan, for financially supporting this research under Contract No. MOST 106-2221-E-119-003 and MOST 107-2410-H-119-005.

**Informed Consent Statement:** All subjects gave their informed consent for inclusion before they participated in the study.

**Data Availability Statement:** Data available on request.

**Conflicts of Interest:** The authors declare no conflict of interest.

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
