# Peer review of "Use of Cloud-Based Virtual Reality in Chinese Glove Puppetry to Preserve Intangible Cultural Heritage"

_applsci, doi:10.3390/app13095699_

Round 1

Reviewer 1 Report

Overall Assessment

The use of digital means and new technologies to give new life to traditional handicrafts and folk art, so that they can better interact with the people, this research is interesting and meaningful. The methodology and technical route adopted in this paper are feasible. The results can be generalized to
a certain extent. It is expected that after the further improvement of this platform, more and more people can participate and experience.

Comments and Suggestions

- It is necessary to clearly state the purpose of the study in chapter I. In particular, how can the use of virtual reality technology change the puppet show? How to achieve the goal of conservation and sustainable development?

- In the second chapter, the author's review of various literatures will help to improve the quality of this manuscript if he can give comments. Otherwise, Chapter 2 seems to be just an enumeration of some literature, what do they inspire for this research? What can be done to improve their shortcomings?

- Is there a chance of breakthrough in the technology used to build this platform? Or does it have new contributions in the process of integrating various technologies? While this can be challenging and difficult, as an applied science journal, the above content is also needed. It is necessary for the authors to explain this.

- The content of the fourth chapter is very specific, which is worthy of recognition! But how does it feel after the user experience? This may be more important. On the one hand, the user's sense of experience determines their acceptance of this folk art. In short, if the user experience is poor,
you may lose interest in the art. On the other hand, the user's experience can in turn be used to continuously improve the mechanisms and models of the platform.

- The second paragraph of the last chapter should be based on the authors' rational analysis of the utility of the platform they have developed. Perhaps this research is just getting started and needs to be improved. But what the second paragraph says, although a beautiful vision, may lack the necessary evidence. It is recommended that the authors adjust the relevant formulation.

- Minor issues: Considering that some readers may not be very familiar with this field, it is recommended to retain the full name when it first appears, such as CCDIK.

Decision: The manuscript needs to be reviewed again after major revisions.

Author Response

  1. It is necessary to clearly state the purpose of the study in chapter I. In particular, how can the use of virtual reality technology change the puppet show? How to achieve the goal of conservation and sustainable development?

 We appreciate the reviewer’s suggestion. We add a paragraph clearly described the purpose of the study in chapter 1 form line 63 to line 81.

The purpose of this study was to use of cloud-based virtual reality in Chinese glove puppetry to preserve intangible cultural heritage. Therefore, this study developed a multiuser cloud-based virtual reality glove puppetry system that enhances the classic works of glove puppetry, as shown in Figure 1. Each user has a unique perception of the virtual environment and can interact remotely. In addition, the Internet enables users to collaborate in different locations to perform together. To ensure that collaboration in VR is effective, participants’ gestures and emotions must be captured and transmitted in real-time [11]. Therefore, a teamwork allows players to work collaboratively way at different places. Finally, a user experience study is included in this article. User experience research focuses on developing a system usability scale and provides an understanding of puppet cultural preservation. The main contributions of this study are as follows:

  1. This paper presents a collaborative work of a puppetry show in a multi-user virtual reality system.
  2. The user experience evaluation was implemented to identify predictive and interpretive use measures for these technologies.
  3. The proposed cloud-based VR system is easy to use and is not limited to users of a particular group.
  4. Our VR system not only provides experience of the operation puppetry but also helps preserve traditional intangible culture.

  1. In the second chapter, the author's review of various literatures will help to improve the quality of this manuscript if he can give comments. Otherwise, Chapter 2 seems to be just an enumeration of some literature, what do they inspire for this research? What can be done to improve their shortcomings?

 We appreciate the reviewer’s comments. Many our opinions of reference papers are added in Chapter 2.

  • from line 173 to line 178.

Unfortunately, they only build a 3D scene with a fixed view point. In fact, their system is a 3D multimedia system without immersion. So far, some above virtual reality systems for single user were established for intangible cultural heritage. No multiuser cloud-based VR systems was developed to present intangible cultural heritage. For this reason, this paper proposed a multiuser cloud-based VR glove puppetry system to protect intangible cultural heritage.

  • from line 192 to line 200.

Some of the above mentioned VR systems have been held back by poor accessibility and lack intuitive multi-user capabilities. MuVR is portable, but has a low resolution. CollaVR is a video-based VR that is realistic, but lacks interaction. Although Dollhouse VR and CoVAR allow users to collaborate on spatial tasks in a shared space, they are not suitable for cultural heritage applications. In fact, the system design and requirements are different depending on the application. Most VR applications require trained people to perform well in a virtual reality environment. Therefore, it is important that user experience studies evaluate system usability and requirements. Unfortunately, most of the aforementioned systems do not conduct user experience studies.

  1. Is there a chance of breakthrough in the technology used to build this platform? Or does it have new contributions in the process of integrating various technologies? While this can be challenging and difficult, as an applied science journal, the above content is also needed. It is necessary for the authors to explain this.

 We appreciate the reviewer’s suggestion. Many descriptions are shown in the revised paper.

  • from line 371 to line 381.

It is more challenging to safeguard intangible cultural heritage than tangible cultural heritage because performance art, music, and practices have no distinct form. This paper describes the requirements for interactive experiences, digitization of glove puppetry, and a VR glove puppetry system. Our glove puppetry design was inspired by Chinese glove puppetry history, which includes one stage and many glove puppets. In the process of implementation, a set of operational procedures for comprising modules, data integration, and technology implementation of virtual reality networks are planned for traditional Chinese glove puppetry. The system enables puppetry to transcend time and space, and becomes a dynamic interactive experience. Users without professional training can control glove puppet avatars using their hands. This sys-tem involves human–computer and human–human interactions and provides an entertaining experience for users of all ages.

  • from line 73 to line 81.

The main contributions of this study are as follows:

  1. This paper presents a collaborative work of a puppetry show in a multi-user virtual reality system.
  2. The user experience evaluation was implemented to identify predictive and interpretive use measures for these technologies.
  3. The proposed cloud-based VR system is easy to use and is not limited to users of a particular group.
  4. Our VR system not only provides experience of the operation puppetry but also helps preserve traditional intangible culture.
  • from line 398 to line 409.

Looking into the practical applications of the future, the VR glove puppetry sys-tem and similar systems can be used to enhance education and increase the accessibility of art forms to the public. VR research also provides the interaction and design for future intergenerational communication for cultural heritage. This system can also be used to promote intangible cultural heritage through interactive VR experiences. It can be used in exhibitions to display intangible cultural heritage, and can be promoted based on numbers to promote the popularization of intangible cultural heritage. Inter-generational interaction and learning using our VR system could lead to an exchange between experience and knowledge of traditional culture. The findings of this study can inform the preservation of this form of traditional art. Thus, cloud-based virtual reality has become accessible and reasonable. Moreover, this study's understandings and accomplishments will deliver and relocate this traditional art form, which makes a new application.

  1. The content of the fourth chapter is very specific, which is worthy of recognition! But how does it feel after the user experience? This may be more important. On the one hand, the user's sense of experience determines their acceptance of this folk art. In short, if the user experience is poor, you may lose interest in the art. On the other hand, the user's experience can in turn be used to continuously improve the mechanisms and models of the platform.

 We appreciate the reviewer’s suggestion. We add a new chapter 5 “user experience study” from line 308 to line 369.

This article presents a study of user experience. In general, the system usability scale [30] is a common reference for usability surveys in case studies for evaluating user experience design. This usability study focuses on conducting user experience research and provides an understanding of the feasible design methodology included in the Chinese puppet cultural preservation VR project. ….

  1. The second paragraph of the last chapter should be based on the authors' rational analysis of the utility of the platform they have developed. Perhaps this research is just getting started and needs to be improved. But what the second paragraph says, although a beautiful vision, may lack the necessary evidence. It is recommended that the authors adjust the relevant formulation.

 Many descriptions are improved according to reviewer’s suggestions. Our analysis of the platform we have developed in the chapter 6 conclusion and future work from line 382 to line 397.

The questionnaire design of this study includes the system usability scale and the Chinese puppet culture 

preservation satisfaction scale. After analysis and statistics, the usability score of this system was 82.08 points, which is higher than the average score of 68 points. This demonstrates that the proposed system has good usability and is easy to use. The satisfaction scores for Chinese puppet culture preservation are high, and our system achieves knowledge sharing and learning regarding cultural inheritance. The system developed in this study contributes substantially to the preservation of glove puppetry.

According to participants’ suggestions, some functions need to be enhanced in the future. For action feedback, many subjects were expected to collide instead of penetration feedback. Collision detection includes collisions between puppets and between puppets and background. In addition to the dialogue each other between users, there should be sound effects or background music. The sound effects showed that the puppet’s performance and experience could adhere to reality. However, the users’ experience cannot be too long because of hand fatigue. Moreover, the subjects were expected to learn from an expert’s puppet skills using our system. In the future, we will provide a performance record system for a grandmaster’s puppet.

  1. Minor issues: Considering that some readers may not be very familiar with this field, it is recommended to retain the full name when it first appears, such as CCDIK.

 All acronyms were indicated when first used.

HTC (High Tech Computer Corporation) in line 149.

VRPN (VR periphery network) in line 270.

CCDIK (Cyclic Coordinate Descent Inverse Kinematics) in line 290.

Reviewer 2 Report

The authors present a very interested approach for protecting intangible cultural heritage. The described a multiuser cloud-based virtual reality glove puppetry platform. The paper is well-written. I suggest some minor improvements like:

 - in the literature research please explain in details [18]. What kind of puppets were used? 

- the title of chapter 3 and 4 should be moved to the next page. 

- the description of the platform is nice. In my opinion the authors also should explain what elements of Chinese culture are protected. What puppets, costumes are used and why (what are they meaning to the culture)?

- in the reference list the following paper have some additional numbers: 4, 5, 6, 7, 12, 14, 15, 16. Please remove them.

Author Response

Reviewer 2:

  1.  in the literature research please explain in details [23]. What kind of puppets were used?

Lin et al. proposed a method for controlling a virtual puppet by using a computer [23]. Users can select from different puppets, scenes, and music. They used Leap Motion controllers to capture the user’s gestures and display the puppet’s movements on a computer screen. Unfortunately, they only build a 3D scene with a fixed view point. In fact, their system is a 3D multimedia system without immersion.

The following two figures were captured from their paper. But we do not have permission to use it.

Figure 2: The members holding puppets to act

Figure 11:Performing the glove puppetry with leap motion

  1. the title of chapter 3 and 4 should be moved to the next page. 

 We appreciate the reviewer’s comments. All of the editing errors are corrected.

  1. the description of the platform is nice. In my opinion the authors also should explain what elements of Chinese culture are protected. What puppets, costumes are used and why (what are they meaning to the culture)?

 We appreciate the editor’s suggestions. It is more challenging to safeguard intangible cultural heritage than tangible cultural heritage because performance art, music, and practices have no distinct form.

  • We exchange section 2.1 and section 2.2. We clearly describe the Chinese glove puppetry developing history in section 2.1 from line 89 to line 128.

Chinese glove puppetry incorporates performing art, visual art, music, classical literature, and folklore. The UNESCO included Chinese glove puppetry in its list of intangible cultural heritage in “Convention for the Safeguarding” in 2012 [12]. Glove puppetry is a type of performance in which puppets are used to animate and create characters, stories, and worlds. .. . .. .

  • Our glove puppetry digital design in chapter 3 was inspired by traditional glove puppetry including stage design (Fig. 4 and Fig. 4) and glove puppet design (Fig. 5) from line 208 to line 239..
  • In order to put it more clearly, we also add our contributions in chapter 1, and provide conclusion and future works in chapter 6.

It is more challenging to safeguard intangible cultural heritage than tangible cultural heritage because performance art, music, and practices have no distinct form. This paper describes the requirements for interactive experiences, digitization of glove puppetry, and a VR glove puppetry system. Our glove puppetry design was inspired by Chinese glove puppetry history, which includes one stage and many glove puppets. In the process of implementation, a set of operational procedures for comprising modules, data integration, and technology implementation of virtual reality networks are planned for traditional Chinese glove puppetry. The system enables puppetry to transcend time and space, and becomes a dynamic interactive experience. Users without professional training can control glove puppet avatars using their hands. This system involves human–computer and human–human interactions and provides an entertaining experience for users of all ages.

From line 395 to line 397 ……. The subjects were expected to learn from an expert’s puppet skills using our system. In the future, we will provide a performance record system for a grandmaster’s puppet.

  1. in the reference list the following paper have some additional numbers: 4, 5, 6, 7, 12, 14, 15, 16. Please remove them.

We appreciate the reviewer’s comments. All of the typing errors are corrected.

Reviewer 3 Report

Dear authors,

The topic addressed is attractive and its approach is clear.

To improve the article, I have a few requests:

Lines 85-89: More details about the motion capture system created and/or bibliographic references to it are needed.

Line 204: The study cannot generate the platform, please rephrase

Lina 252 (before the Conclusions): 

-Clarifications are needed regarding the appeal of the virtual reality glove puppetry platform. Why would users choose this platform over others?

-It would also be useful to further emphasize the cultural heritage value of glove puppetry.

Please indicate the meaning of all acronyms when first used (HTC, HMD, VRPN, CCDIK, etc.)

Please consider these requests constructive. Success!

Author Response

Reviewer 3:

  1. Lines 85-89: More details about the motion capture system created and/or bibliographic references to it are needed.

 We appreciate the reviewer’s suggestion. This is our previous work. The details description is shown in the revised paper from line 146 to line 155.

Further, Day and Way established a VR program for a white crane folk dance performed in Mazu temples in Taiwan [18], as shown in Figure 4. In first phase, the Kinect module was used to capture the performance of the folk dancers and animates the motion to the 3D animation characters (white crane). In second phase, the Kinect and HTC VIVE tracker were used to capture user actions. Through the Unity integration platform, the 3D animation characters played and the 3D avatar to be instantly driven were presented to user in the environment. A VR headset was paired with the Kinect motion capture sensor to immerse users into a 3D virtual world where they can view the dancing cranes and appreciate a dynamic virtual environment in real time.

The following figures about motion capture system was displayed in our original paper.

System Architecture of White Crane Dance VR using Kinect and HTC with Unity

  1. Line 204: The study cannot generate the platform, please rephrase

 We appreciate the reviewer’s suggestion. We replaced “Cloud-based Virtual Reality platform” with “Cloud-based Virtual Reality System”.

  1. Lina 252 (before the Conclusions): 

-Clarifications are needed regarding the appeal of the virtual reality glove puppetry platform. Why would users choose this platform over others?

 We appreciate the reviewer’s suggestion. We add a new chapter 5 “user experience study” from line 308 to line 369.

This article presents a study of user experience. In general, the system usability scale [30] is a common reference for usability surveys in case studies for evaluating user experience design. This usability study focuses on conducting user experience research and provides an understanding of the feasible design methodology included in the Chinese puppet cultural preservation VR project. ….

  1. -It would also be useful to further emphasize the cultural heritage value of glove puppetry.

We appreciate the editor’s suggestions. It is more challenging to safeguard intangible cultural heritage than tangible cultural heritage because performance art, music, and practices have no distinct form.

  • We exchange section 2.1 and section 2.2. We clearly describe the Chinese glove puppetry developing history in section 2.1 from line 89 to line 128.

Chinese glove puppetry incorporates performing art, visual art, music, classical literature, and folklore. The UNESCO included Chinese glove puppetry in its list of intangible cultural heritage in “Convention for the Safeguarding” in 2012 [12]. Glove puppetry is a type of performance in which puppets are used to animate and create characters, stories, and worlds. .. . .. .

  • Our glove puppetry digital design in chapter 3 was inspired by traditional glove puppetry including stage design (Fig. 4 and Fig. 4) and glove puppet design (Fig. 5) from line 208 to line 239..
  • In order to put it more clearly, we also add our contributions in chapter 1, and provide conclusion and future works in chapter 6.

It is more challenging to safeguard intangible cultural heritage than tangible cultural heritage because performance art, music, and practices have no distinct form. This paper describes the requirements for interactive experiences, digitization of glove puppetry, and a VR glove puppetry system. Our glove puppetry design was inspired by Chinese glove puppetry history, which includes one stage and many glove puppets. In the process of implementation, a set of operational procedures for comprising modules, data integration, and technology implementation of virtual reality networks are planned for traditional Chinese glove puppetry. The system enables puppetry to transcend time and space, and becomes a dynamic interactive experience. Users without professional training can control glove puppet avatars using their hands. This system involves human–computer and human–human interactions and provides an entertaining experience for users of all ages.

From line 395 to line 397 ……. The subjects were expected to learn from an expert’s puppet skills using our system. In the future, we will provide a performance record system for a grandmaster’s puppet.

  1. Please indicate the meaning of all acronyms when first used (HTC, HMD, VRPN, CCDIK, etc.)

All acronyms were indicated when first used.

HTC (High Tech Computer Corporation) in line 149.

We replaced HMD with “head-mounted display” in line 261 and line 322.

VRPN (VR periphery network) in line 270.

CCDIK (Cyclic Coordinate Descent Inverse Kinematics) in line 290.

Round 2

Reviewer 1 Report

Thnaks for the authors made the efforts to review this amnuscript.

Author Response

We appreciate the reviewer’s comments. We are glad to learn a lot of from the reviewers who made many excellent comments to improve our paper presentation. 

Reviewer 3 Report

Dear authors,

I still think it would be helpful to highlight

Why would users choose this platform over others?.

Author Response

We appreciate the reviewer’s comments. We are glad to learn a lot of from the reviewers who made many excellent comments to improve our paper presentation.

 I still think it would be helpful to highlight. Why would users choose this platform over others?

  •  The purpose of this study in chapter 1 form line 64 to line 81.

This study developed a multiuser cloud-based virtual reality glove puppetry system that enhances the classic works of glove puppetry, as shown in Figure 1. Each user has a unique perception of the virtual environment and can interact remotely. In addition, the Internet enables users to collaborate in different locations to perform together. To ensure that collaboration in VR is effective, participants’ gestures and emotions must be captured and transmitted in real-time [11]. Therefore, a teamwork allows players to work collaboratively way at different places. Finally, a user experience study is included in this article. User experience research focuses on developing a system usability scale and provides an understanding of puppet cultural preservation. The main contributions of this study are as follows:

  1. This paper presents a collaborative work of a puppetry show in a multi-user virtual reality system.
  2. The user experience evaluation was implemented to identify predictive and interpretive use measures for these technologies.
  3. The proposed cloud-based VR system is easy to use and is not limited to users of a particular group.
  4. Our VR system not only provides experience of the operation puppetry but also helps preserve traditional intangible culture.

  • Many our opinions of reference papers are added in Chapter 2.

from line 167 to line 178.

Nitsche and McBride explored controlling puppets in VR and puppet-based VR games [22]. Through their bottom-up design approach, they created a VR game in which users adopt the puppets perspective, thereby demonstrating the various design possibilities that VR offers. Lin et al. proposed a method for controlling a virtual pup-pet by using a computer [23]. Users can select from different puppets, scenes, and music. They used Leap motion controllers to capture the user’s gestures and display the puppet’s movements on a computer screen. Unfortunately, they only build a 3D scene with a fixed view point. In fact, their system is a 3D multimedia system without immersion. So far, some above virtual reality systems for single user were established for intangible cultural heritage. No multiuser cloud-based VR systems was developed to present intangible cultural heritage. For this reason, this paper proposed a multiuser cloud-based VR glove puppetry system to protect intangible cultural heritage.

from line 192 to line 200.

Some of the above mentioned VR systems have been held back by poor accessibility and lack intuitive multi-user capabilities. MuVR is portable, but has a low resolution. CollaVR is a video-based VR that is realistic, but lacks interaction. Although Dollhouse VR and CoVAR allow users to collaborate on spatial tasks in a shared space, they are not suitable for cultural heritage applications. In fact, the system design and requirements are different depending on the application. Most VR applications require trained people to perform well in a virtual reality environment. Therefore, it is important that user experience studies evaluate system usability and requirements. Unfortunately, most of the aforementioned systems do not conduct user experience studies.

  • Result of the Chinese puppet culture preservation questionnaire in Chapter 5 from line 355 to line 367.

Table 4 shows the results of the survey questions on Chinese puppet culture preservation. The scores for all the items were above 4.3. The proposed cloud-based VR system not only provides learning and experience of the operation and performance of puppetry, but also helps to preserve traditional intangible culture. Specially, cloud-based VR brings new cultural experiences or knowledge learning in the post-COVID-19 era.

Furthermore, we asked participants for their comments. On the positive side, users said they liked "The technology is very interesting, fancy and impressive.”, "It gives me a good visual immersive experience.” and "It is great for people new to VR. It has been a great experience." Users also indicated that they desire for “Lacks the tactile feel of puppet physical objects.”, “Problem of force feedback during puppet fighting.”,” Although there are dialogues, it would be better to play sound effects or background music.”,” The user would like to learn from an expert puppeteer.”

  • Conclusion and future works in chapter 6 from line 371 to line 397.

It is more challenging to safeguard intangible cultural heritage than tangible cultural heritage because performance art, music, and practices have no distinct form. This paper describes the requirements for interactive experiences, digitization of glove puppetry, and a VR glove puppetry system. Our glove puppetry design was inspired by Chinese glove puppetry history, which includes one stage and many glove puppets. In the process of implementation, a set of operational procedures for comprising modules, data integration, and technology implementation of virtual reality networks are planned for traditional Chinese glove puppetry. The system enables puppetry to transcend time and space, and becomes a dynamic interactive experience. Users without professional training can control glove puppet avatars using their hands. This system involves human–computer and human–human interactions and provides an entertaining experience for users of all ages. The questionnaire design of this study includes the system usability scale and the Chinese puppet culture preservation satisfaction scale. After analysis and statistics, the usability score of this system was 82.08 points, which is higher than the average score of 68 points. This demonstrates that the proposed system has good usability and is easy to use. The satisfaction scores for Chinese puppet culture preservation are high, and our system achieves knowledge sharing and learning regarding cultural inheritance. The system developed in this study contributes substantially to the preservation of glove puppetry.
